# EEG-MACS: Manifold Attention and Confidence Stratification for EEG-based Cross-Center Brain Disease Diagnosis under Unreliable Annotations

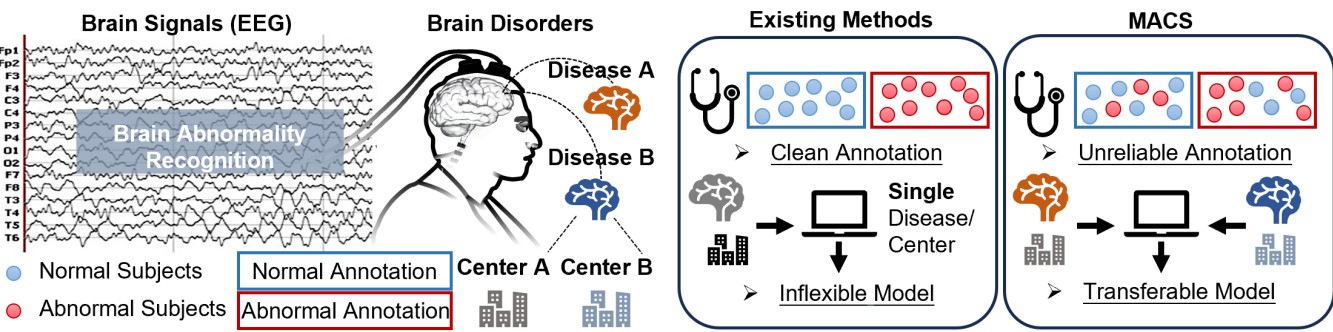

**Figure 1: The proposed MACS framework tackles data heterogeneity and annotation unreliability in EEG-based brain disease diagnosis, aiming for superior performance across centers and validation in both neurocognitive and movement disorders.**

## ABSTRACT

Cross-center data heterogeneity and annotation unreliability significantly challenge the intelligent diagnosis of diseases using brain signals. A notable example is the EEG-based diagnosis of neurodegenerative diseases, which features subtler abnormal neural dynamics typically observed in small-group settings. To advance this area, in this work, we introduce a transferable framework employing **M**anifold **A**ttention and **C**onfidence **S**tratification (**MACS**) to diagnose neurodegenerative disorders based on EEG signals sourced from four centers with unreliable annotations. The MACS framework's effectiveness stems from these features: 1) The ***Augmentor*** generates various EEG-represented brain variants to enrich the data space; 2) The ***Switcher*** enhances the feature space for trusted samples and reduces overfitting on incorrectly labeled samples; 3) The ***Encoder*** uses the Riemannian manifold and Euclidean metrics to capture spatiotemporal variations and dynamic synchronization in EEG; 4) The ***Projector***, equipped with dual heads, monitors consistency across multiple brain variants and ensures diagnostic accuracy; 5) The ***Stratifier*** adaptively stratifies learned samples by confidence levels throughout the training process; 6) Forward and backpropagation in **MACS** are constrained by confidence stratification to stabilize the learning system amid unreliable annotations. Our subject-independent experiments, conducted on both neurocognitive and movement disorders using cross-center corpora, have

demonstrated superior performance compared to existing related algorithms. This work not only improves EEG-based diagnostics for cross-center and small-setting brain diseases but also offers insights into extending MACS techniques to other data analyses, tackling data heterogeneity and annotation unreliability in multimedia and multimodal content understanding. We have released our code here: https://anonymous.4open.science/r/EEG-Disease-MACS-0B4A.

## CCS CONCEPTS

• **Computing methodologies** → **Modeling methodologies**; • **Applied computing** → *Health care information systems*; • **Human-centered computing**;

## KEYWORDS

EEG Signals, Cross-center Learning, Unreliable Annotation, Weakly-supervised Learning, Neurodegenerative Disease

## 1 INTRODUCTION

Addressing the challenges posed by data heterogeneity and annotation unreliability is essential for the analysis and interpretation of multimedia and multimodal data [1, 2]. These issues are particularly prevalent in human-centered computing, where they encounter even greater difficulties in small-group settings. A prime example is the analysis of neural dynamics, such as the automated diagnosis of cognitive and movement brain disorders through electroencephalogram (EEG) signals [3, 4]. While EEG-based diagnosis models have undergone extensive research, they are often disease-specific, rely on single-center studies, and depend on fully supervised learning with annotated data [5]. Developing effective and transferable models capable of handling cross-center data and low-quality annotations is crucial for enhancing adaptability across different diseases and advancing EEG modeling techniques.

**Unpublished working draft. Not for distribution.**

Current EEG analyzing methods range from traditional feature engineering to deep learning approaches. Various features extracted from domains like time (e.g., Hjorth parameters), frequency (e.g., Fourier Transform), dynamics (e.g., entropy), and functional networks (e.g., synchronization and phase coupling) have demonstrated efficacy due to EEG's intrinsic characteristics [6–9]. The sparsity of handcrafted features has led to the adoption of deep learning models, such as convolutional neural networks, graph neural networks, and hybrid approaches, to capture spatial-temporal patterns, structured characteristics, and other high-level representations [10–13]. However, these existing methods often struggle with noisy data and lack cross-center applicability. This may primarily be due to: **1)** *the ineffective mapping that cannot project EEG signals into a high-level feature space invariant across diverse data distributions from multiple centers;* **2)** *the absence of integrating advanced learning strategies to tackle complex learning issues under unreliable annotations effectively.*

From this perspective, our work presents a novel framework aimed at developing a transferable model adept at managing unreliable annotations, as illustrated in Figure 1. This model specifically targets the recognition of neurodegenerative disorders. These include neurocognitive disorders, represented by mild cognitive impairment(MCI) and Alzheimer's Disease(AD), and movement disorders, exemplified by Parkinson's Disease(PD). We selected these three diseases due to the potential progression relationship between AD and MCI [14] as well as the observed comorbidity between PD and cognitive impairment [15, 16]. The proposed framework is characterized by its innovative use of **M**anifold **A**ttention and **C**onfidence **S**tratification (MACS), which optimally synergizes modules including the *Augmentor, Switcher, Encoder, Projector,* and *Stratifier.* MACS's success in achieving transferability across centers and diseases under unreliable annotations can be attributed to three key factors: **1)** *The establishment of an optimized mapping that leverages the strengths of both Euclidean and Riemannian manifold spaces enables the model to extract more effective EEG representations.* **2)** *The integration of supervised and self-supervised learning strategies through confidence stratification to address the issue of learning from unreliable annotations.* **3)** *An effective encoder enhances the representation space, which improves confidence stratification accuracy, thereby enabling the encoder to develop more robust and inductive representations in a mutually reinforcing cycle.* Our primary contributions are outlined below:

- Introducing a novel EEG-based framework, MACS, designed for learning from unreliable annotated EEG signals with cross-center transferability.
  - ***Augmentor*:** Enriches the data space via augmentation.
  - ***Switcher*:** Mitigates overfitting on incorrect labels.
  - ***Encoder*:** Integrates Manifold and Euclidean spaces to improve EEG representation learning.
  - ***Projector*:** Features dual heads to assess both data consistency and diagnostic accuracy.
  - ***Stratifier*:** Stratifies data based on confidence levels.
- Establishing confidence stratification-based constraints for MACS's forward and back-propagation that create a self-organizing system to enhance representation learning and confidence assessment, thereby fostering a virtuous cycle.

- Demonstrating MACS's superior performance in learning from unreliable annotations compared to state-of-the-art (SOTA) methods, validated across two types of diseases and through cross-center testing and fine-tuning.
- Making the code public to contribute multimedia community.

## 2 RELATED WORK

### 2.1 Learning with Unreliable Annotations

Noisy label learning contains model-free and model-based strategies aimed at reducing the influence of incorrect labels. These strategies involve estimating noise patterns and conducting supervised learning with clean samples [17], as well as handling noisy labels and refining the model by addressing internal conflicts [18, 19]. Such methods depend on the analysis of the relationships within noisy data. Recent research in time-series [20] and image domains [21, 22] applies unique principles to delineate relationships. In the image domain, Promix [21] adopts small-loss criteria and prediction consistency to filter high-confidence examples, followed by learning through Debiased Semi-Supervised Training. Sel-CL [22] employs nearest neighbors to select confident pairs for supervised contrastive learning. In the time-series domain, CTW [20] selects confident examples based on small-loss criteria and applies time-warping to these instances to learn more robust representations. We use these to benchmark MACS and, following [23, 24], incorporate mix-up techniques [25, 26] to enhance learning from noisy labels.

### 2.2 Contrastive Learning for Time Series

Contrastive learning has demonstrated effectiveness in both unsupervised (e.g., SimCLR [27]) and supervised (e.g., SupCon [28]) contexts. It provides valuable self-learning strategies for pre-training in time series data, such as enhancing time-frequency [29] and cross-sample temporal consistency [3]. However, the impact of consistency pre-training on handling unreliable annotated data is less pronounced. Augmentation-based contrast [30] shows promise for semi-supervised learning with missing labels but is less effective without true label priors. To overcome these limitations, MACS introduces multi-view contrastive learning, guided by estimated confidence levels.

### 2.3 Manifold Learning for EEG Signals

Manifold-based modeling excels in EEG signal analysis for brain-computer interfaces (BCI) [31, 32], leveraging Riemannian geometry for high-dimensional neural data representation through affine-invariant geometric distances [33, 34]. Using EEG's Riemannian structure for BCI domain-adaptation [10, 35, 36] and multi-task application [11] indicates the potential of Manifold geometry learning in identifying robust latent spaces. Such evidence bolsters the *Encoder*'s design in MACS. Our work advances current geometry learning by integrating dynamic functional networks with cross-attention mechanisms. Additionally, findings from our pilot study suggest that when integrated with contrastive learning, Riemannian geometry learning offers a promising alternative to GNNs [37].

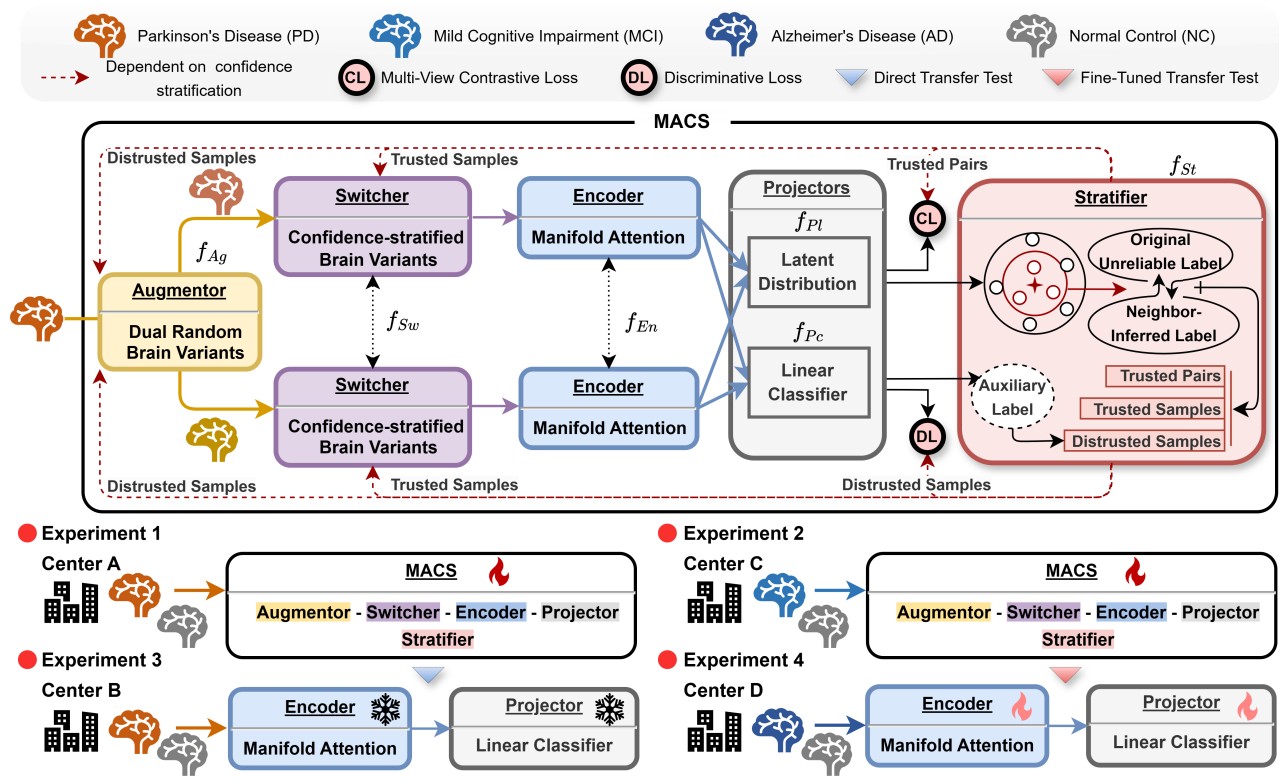

Figure 2: Overview of the MACS framework and subject-independent experiments across diseases and centers. The *Projector* has a dual-head structure for latent space representation and classification. The *Stratifier* categorizes samples by label confidence, constraining brain variants for contrastive learning and discriminative loss for distrusted samples. Refer to Figures 3 and 4 for *Switcher* and *Encoder*, respectively.

## 3 MACS

In this section, we elucidate the methodology for learning from brain signals annotated with unreliable labels using the Manifold Attention and Confidence Stratification (MACS) framework. As depicted in Figure 2, MACS handles unreliable labels and enhances cross-center transferability through its modules (in *Section* 3.1), adaptive constraints (in *Section* 3.2), and training objective(in *Section* 3.3).

Assuming that brain activity is monitored using $d$ sensors at a sampling frequency of $f_s$ for a duration of $t$ seconds, we obtain the following set of observations as the input for the MACS framework:

$$A_{d,T}^{(n)} = \begin{pmatrix} a_{1,1} & a_{1,2} & \cdots & a_{1,T} \\ a_{2,1} & a_{2,2} & \cdots & a_{2,T} \\ \vdots & \vdots & \ddots & \vdots \\ a_{d,1} & a_{d,2} & \cdots & a_{d,T} \end{pmatrix}, \quad (1)$$

where $T = f_s \cdot t$ denotes the total number of sampling points, and $n \in \{1, 2, \ldots, N\}$ represents a specific individual from a total of $N$ subjects under observation.

The MACS framework identifies brain states as $\hat{Y}^{(n)} = F_\Theta(A_{d,T}^{(n)})$ in scenarios with unreliable annotations $Y^{(n)}$, utilizing five key components: 1) *Augmentor*, 2) *Switcher*, 3) *Encoder*, 4) *Projector*, and

5) *Stratifier*. The initial four modules are integral to the framework's forward propagation, while the *Stratifier* systematically regulates the learning mechanism.

### 3.1 MACS Modules

*Augmentor: Producing Brain Variants through Dual Random Transform.* In this module, data augmentation is applied to the preprocessed brain data as described in [29], following the equation:

$$f_{Ag}(A_{d,T_s}, \sigma) = A_{d,T_s} + \epsilon, \quad (2)$$

where $\epsilon_{ij} \sim \mathcal{N}(0, \sigma^2)$ is independently and identically distributed, with $\sigma$ representing the standard deviation. Additionally, $A_{d,T_s}$ is a segmented, non-overlapping fragment of $A_{d,T}$, utilized to reduce computational load and increase the system's robustness. Two random transformations are concurrently executed, resulting in dual brain variants. Differing from the method of specific weak and strong augmentations for two views [38], our approach operates a random configuration, leading to improved performance.

*Switcher: Blending-based Variant Generation Conditional on Confidence Levels.* In the *Switcher* module, dual brain variants undergo selective processing. As in Figure 3, distrusted inputs are directly forwarded to subsequent modules, while trusted ones are routed to

a blender for generating interpolated samples. Inspired by the benefits of sample and network mixing, as discussed in [25] and [23], and aiming to enhance representation learning with partial labels and reduce confirmation bias in noisy data, we distort each trusted sample $A^*_{d,T_s}$ and its corresponding label $y^*$ by interpolating it with another randomly selected sample $(A^+_{d,T_s}, y^+)$, to produce blended variants, as described in Eq.(3). $\lambda$ is sampled from a uniform Beta(1, 1) distribution and adjusted to ensure it is always at least 0.5 by setting it to the greater value between $\lambda$ and $1 - \lambda$.

$$f_{Sw}((A^*_{d,T_s}, y^*), \lambda) = \lambda(A^*_{d,T_s}, y^*) + (1 - \lambda)(A^+_{d,T_s}, y^+) \quad (3)$$

This choice is implemented to balance the retention of the primary input's information and the introduction of diversity from the randomly indexed input throughout the mixing process.

Differing from the referenced studies [23, 25], our method presents two distinctions: 1) We produce dual-blended brain variants while employing shared network modules, thus effectively decreasing model complexity. 2) We initiate data blending exclusively on samples exhibiting higher confidence levels, aiming to mitigate concerns related to unreliable labels. The confidence level associated with each sample in the mini-batch is evaluated using the *Stratifier* module in the MACS framework.

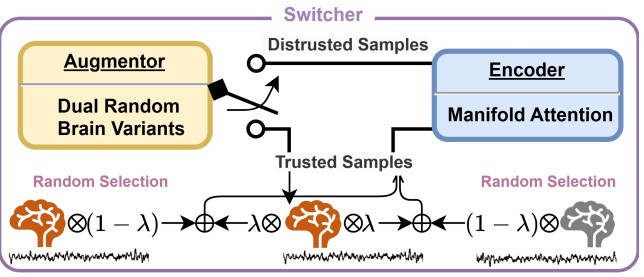

**Figure 3: The *Switcher* employs conditional interpolated blending for trusted samples and bypasses distrusted ones to mitigate overfitting on incorrectly labeled samples.**

*Encoder: Mapping EEG Dynamics onto Riemannian Manifold for Attention-Based Analysis.* The *Encoder* module $f_{En}$, grounded in our feature engineering findings (*Section* 4.3), initiates with a convolution starter $g_{str}$. This component reduces noise/artifacts and captures specific wave patterns (e.g., theta, alpha rhythms) and time-domain features of brain activity using convolutions and normalization, effectively decoding brain signals at a lower level[39]. This process yields $X_{d,T_s} = g_{str}(A_{d,T_s})$. Subsequently, a temporal clipper $g_{clp}$ is applied to these low-level features $X_{d,T_s}$ across the sampling dimension. This operation enhances their representation by leveraging the high temporal resolution of EEG signals. A series of feature clips $\{X_{dim_1, t_i}\}_{i=1}^I$ are then processed by a synchronization analyzer $g_{syn}$, which measures the coupling between pairs of $d$ sensors, ultimately producing a dynamic functional network represented as $\Phi = [\phi_{t_1}, \dots, \phi_{t_i}, \dots, \phi_{t_I}]$.

With evidence from learning Riemannian geometry for structured brain data as shown in [11, 36], a manifold converter $g_{mfd}$

transforms dynamic network $\Phi$ into a set of symmetric positive definite (SPD) matrices, $S_d^{++}(\mathbb{R})$, thus defining a Riemannian manifold $\mathcal{M}$. This transformation involves eigenvalue decomposition and transpose manipulation of each adjacency matrix, represented as $\mathcal{D} = g_{mfd}(\Phi)$. To measure those $S_d^{++}$, we utilized the Log-Euclidean Riemannian metric [33], as it serves as the first-order approximation of the Affine-Invariant metric. At each point $\mathcal{D}_{t_i}$ on the manifold $\mathcal{M}$, a tangent space is defined by $\log : \mathcal{M} \to \mathcal{T}_\mathcal{M}$, where the inverse is operated by exponential mapping, as illustrated in Figure 4. The distance between any two points on the manifold $\mathcal{M}$ is then calculated using the following equation:

$$d(\mathcal{D}_{t_i}, \mathcal{D}_{t_j}) = \left\| \log(\mathcal{D}_{t_i}) - \log(\mathcal{D}_{t_j}) \right\|_F^2. \quad (4)$$

Utilizing this metric to reflect the correlation between points $\mathcal{D}_{t_i} \in S_d^{++}$, we constructed a manifold-based dynamic attention block $g_{datt}$, following the approach in [11], but with modifications to incorporate a cross-temporal attention mechanism. This integration of dynamic relationships is mathematically represented as:

$$\mathcal{F}_i = \sum_{j=1, j \neq i}^I \delta \left( d(f_{W_k}(\mathcal{D}_{t_i}), f_{W_q}(\mathcal{D}_{t_j})) \right) \cdot \log(f_{W_v}(\mathcal{D}_{t_i})), \quad (5)$$

where $\delta$ scales the values to a range of 0 to 1. $f_{W_k}$ represents the operation of $W_k$ multiplied by a matrix and then by $W_k^\top$. This can be expressed as $f_{W_k}(X) = W_k X W_k^\top$. Similarly, $f_{W_q}$ and $f_{W_v}$ operate in the same manner, where $W_q$, $W_k$, and $W_v \in \mathbb{R}^{d \times d_1}$ are learned weight matrices used for the bilinear mapping. Subsequently, the fused feature matrices $\mathcal{F} = \{\mathcal{F}_i\}$ are subjected to a nonlinear activation process, involving the rectification of eigenvalues, as detailed in [33]. To enable the measurement of these features $\mathcal{F}$ in Euclidean space for further processing, the embeddings $\hat{\mathcal{F}}$ are obtained by mapping $\mathcal{F}$ onto the tangent space, followed by flattening and concatenating them together.

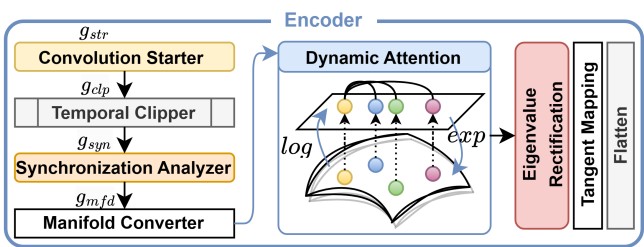

**Figure 4: The *Encoder* combines Riemannian and Euclidean metrics for feature extraction, leading to a manifold-based attention mechanism that effectively captures characteristics of spatiotemporal complexity and dynamic synchronization.**

*Projector: Dual-Branch Mapping for Latent Distribution and Discriminative Classification.* The high-level features encoded by $f_{En}$ are fed into a dual-branch projector. In this setup, one branch, $f_{Pl}$, is dedicated to reducing the feature dimension for confidence estimation and contrastive learning, obtaining $\mathcal{Z} = f_{Pl}(\hat{\mathcal{F}})$, while the other branch, $f_{Pc}$, functions as a classifier for discrimination purposes, obtaining $\mathcal{E} = f_{Pc}(\hat{\mathcal{F}})$.

*Stratifier: Measuring Latent Similarity for Confidence Stratification.* This module $f_{St}$ plays a crucial role in assessing the reliability of the learned representations, which in turn influences the learning mechanism of the MACS framework. It categorizes the confidence levels of samples based on $\mathcal{Z}$ through the following steps:

- Measuring the cosine similarity between samples in $\mathcal{Z}$.

$$c\left(z_i, z_j\right) = \frac{z_i z_j^\top}{\|z_i\|\,\|z_j\|} \tag{6}$$

- Locating the K closet neighbors of each sample $z_i$ in $\mathcal{Z}$ using the k-Nearest Neighbors method.
- Obtaining a neighbor-determined label $\bar{y}$ for each sample $z_i$ by averaging the original labels of its neighbors.

$$p_c\left(z_i\right) = \frac{1}{K} \sum_{\substack{k=1 \\ z_k \in \mathcal{N}_i}}^{K} \mathbb{1}_{y_k=c}, c \in \{0, 1\},$$
$$\bar{y} = \arg\max_c p_c(z_i), \tag{7}$$

where $\mathcal{N}_i$ denotes the neighbourhood of K samples to $z_i$. The indicator function, denoted as $\mathbb{1}_B$, is a function that returns 1 when the condition $B$ is satisfied and 0 otherwise.

- Considering a sample trusted if its neighbor-determined label matches its original label. Following [40], we also adopted a dynamic thresholding approach to ensure class balance.
- Identifying trusted pairs $\Psi$ from trusted samples $\mathcal{Z}_{\mathrm{crd}}$ based on label identity.

For the distrusted samples $\mathcal{Z}_{\mathrm{dst}} = \mathcal{Z} \setminus \mathcal{Z}_{\mathrm{crd}}$, corresponding auxiliary labels are generated through the forward-propagation process $A \xrightarrow{f_{En}} \hat{\mathcal{F}} \xrightarrow{f_{Pc}} \hat{Y}$.

## 3.2 MACS Constraints

MACS constraints are dependent on the confidence stratification adaptively assessed by the *Stratifier* module during training, which is pivotal in stabilizing representation learning in the context of unreliably annotated data. Intuitively, they regulate the behavior of the modules, influencing the forward propagation of features and the computation of backward losses, as depicted in Figure 2. Four types of constraints are implemented within the framework: they are applied to both the *Augmentor* and *Switcher* modules, and also influence the multi-view contrastive loss and discriminative loss. Those constraints work as follows:

1) The first constraint specifically controls the gradients of *Switcher* module $\nabla f_{Sw}$. It effectively utilizes linearly interpolated samples to enrich data associated with trusted labels while simultaneously mitigating the risk of being misled by unreliable data.

2) The second constraint type facilitates a 'divide and conquer' approach within the multi-view contrastive loss. Specifically, contrastive learning is initiated within three groupings: among trusted pairs $\Psi$, among blended variants of trusted samples $\mathcal{Z}_{\mathrm{crd}}$, and among dual random brain variants $\mathcal{Z}$.

3) The third constraint directs the *Augmentor* module $f_{Ag}$ to bypass distrusted samples and put them into the *Encoder* and *Projector*

---

**Algorithm 1** MACS

**Input:** $A_{d,T}$ with unreliable annotation $Y$, maximum epochs $T_{\max}$
**Output:** Learned Model $\Theta$

1: $A_{d,T}$ is segmented into non-overlapping fragments $A_{d,T_s}$
2: **for** $t = 1, 2, \ldots, T_{\max}$ **do**
3:     *Stratifier*: Identify trusted examples $A_{\mathrm{tru}}$ and distrusted examples $A_{\mathrm{dst}}$ based on the consistency between $\mathcal{Z} = f_{Pl}(f_{En}(A_{d,T_s}))$ and provided unreliable labels $Y$
4:     *Augmentor*: $f_{Ag}(A_{d,T_s}, \sigma) = A_{d,T_s} + \epsilon$, generate two brain variants
5:     **if** samples $\in A_{\mathrm{dst}}$ **then**
6:         *Encoder*: $\hat{\mathcal{F}} = f_{En}(A_{d,T_s})$
7:         *Projector*: $\mathcal{Z} = f_{Pl}(\hat{\mathcal{F}}), \mathcal{E} = f_{Pc}(\hat{\mathcal{F}})$
8:         **Loss:** Compute self-supervised contrastive loss $\mathcal{L}^{Ag}$ and use softmax prediction $\hat{Y}$ without data augmentation to compute discriminative loss $\mathcal{L}^{DL}$
9:     **else**
10:         *Switcher*: $f_{Sw}((A_{d,T_s}^*, y^*), \lambda) = \lambda(A_{d,T_s}^*, y^*) + (1 - \lambda)(A_{d,T_s}^+, y^+)$
11:         *Encoder*: $\hat{\mathcal{F}} = f_{En}(A_{d,T_s})$
12:         *Projector*: $\mathcal{Z} = f_{Pl}(\hat{\mathcal{F}}), \mathcal{E} = f_{Pc}(\hat{\mathcal{F}})$
13:         **Loss:** Compute mixed self-supervised loss $\mathcal{L}^{Sw}$, supervised contrastive loss $\mathcal{L}^{St}$, and use original label $Y$ without data augmentation to compute discriminative loss $\mathcal{L}^{DL}$
14:     **end if**
15: **end for**

---

to generate auxiliary labels, providing alternatives to unreliable labels for backpropagation.

4) The fourth constraint conditionally influences the discriminative loss. It evaluates the errors of trusted samples using their original labels, while the gradients for distrusted samples are computed based on auxiliary labels.

## 3.3 MACS Training Objective

The overall training objective, as formulated in Eq.(8), is an equal sum of contrastive and discriminative losses.

$$\mathcal{L} = \mathcal{L}^{CL} + \mathcal{L}^{DL} \tag{8}$$

### 3.3.1 *Multi-view Contrastive Loss.*
In our framework, the contrastive loss $\mathcal{L}^{CL}$ consists of three components: $\mathcal{L}^{Ag}$, $\mathcal{L}^{Sw}$, and $\mathcal{L}^{St}$, each addressing different aspects of the data. All components adhere to the core formulas defined in Eq.(9). The computation involves aggregating $C_{i,j}$ for all pairs of samples $x_j$ within the same minibatch that share the same label as $x_i$ (i.e., $y_i = y_j$), across all $N_{y_i}$ such samples. Each minibatch contains $N_m$ samples, excluding self-contrast cases where $i = j$. A temperature parameter $\tau$ modulates the scale of the dot products in the softmax denominator.

$$\mathcal{L}_i^O(z_i, y_i) = -\frac{1}{2N_{y_i} - 1} \sum_{j=1}^{2N_m} \mathbb{1}_{i \neq j} \mathbb{1}_{y_i = y_j} C_{ij},$$
$$C_{ij} = \log\left(\frac{\exp(z_i \cdot z_j / \tau)}{\sum_{r \in \{1, \ldots, 2N_m\} \setminus \{i\}} \exp(z_i \cdot z_r / \tau)}\right). \tag{9}$$

1) $\mathcal{L}^{Ag}$: It aims to minimize the distance between dual augmentations for distrusted examples $\mathcal{Z}_{dst}$ , which constitutes a self-supervised learning approach.

2) $\mathcal{L}^{Sw}$: This targets the enhancement of similarity among blending variants, but exclusively for trusted samples $\mathcal{Z}_{crd}$, which constitutes a mixed self-supervised learning approach.

3) $\mathcal{L}^{St}$: This component incorporates interpolation supervised contrastive learning techniques, as exploited in [22, 40], and shares the same hyperparameter $\lambda$ with Eq.(3). It effectively combines $\mathcal{L}_{i \in \Psi^*}^{St}$ and $\mathcal{L}_{i \in \Psi^+}^{St}$ from trusted pairs $\Psi = \Psi^* \cup \Psi^+$. It can be expressed as $\mathcal{L}_i^{St} = \lambda \mathcal{L}_i\left(z_i, y^*\right) + (1 - \lambda)\mathcal{L}_i\left(z_i, y^+\right)$.

*3.3.2 Discriminative Loss.* This loss function plays a direct role in quantifying the recognition of abnormalities. It is conditionally structured as detailed in Eq.(10). In this equation, $Y^{(n)}$ represents the original label, $\hat{Y}^{(n)}$ denotes the generated auxiliary label. Algorithm 1 presents the pseudocode for MACS.

$$
\begin{aligned}
\mathcal{L}_i^{DL} = &- \lambda \tilde{Y}^{(*)T} \log(\mathcal{E}^{(i)}) \\
&- (1 - \lambda)\tilde{Y}^{(+)T} \log(f_{Pc}(\mathcal{E}^{(i)})), \\
\tilde{Y}^{(n)} = &\begin{cases} Y^{(n)}, & \text{if } A_{d,T_s}^{(n)} \in A_{crd}, \\ \hat{Y}^{(n)}, & \text{if } A_{d,T_s}^{(n)} \in A_{dst}. \end{cases}
\end{aligned}
\tag{10}
$$

## 4 EXPERIMENTS

We conduct four subject-independent experiments (presented in Figure 2) to validate MACS's performance in *learning from unreliable annotations* and *transferability*: 1) Three-fold cross-validation using public PD data from Center A (Model $\Theta_{PD}$); 2) Four-fold cross-validation with MCI (MCI due to AD) data collected from Center C (Model $\Theta_{MCI}$); 3) Direct testing $\Theta_{PD}$ on a cross-center PD corpus (Center B) for transferability evaluation; 4) Fine-tuning $\Theta_{MCI}$ on a cross-center AD corpus (Center D) for transferability assessment. Please refer to **Appendix** for implementation details.

### 4.1 Experimental Datasets

*PD Datasets.* The PD data includes two sets. The first set, from the University of New Mexico(NMU), comprises data from 27 patients and 27 controls [7]. The second set, from the University of Iowa(IU), includes data from 14 patients and 14 controls [42]. Both datasets were acquired using 64-channel Ag/AgCl electrodes with the Brain Vision system at a sampling rate of 500 Hz.

*MCI Dataset.* The MCI dataset, from a local hospital in City C, includes data from 46 MCI patients and 43 age-matched normal controls (NC), recorded in an eye-closed state. Data was captured using a 64-channel Ag/AgCl electrodes Brain Product system at a sampling rate of 5000 Hz.

*AD Dataset.* The AD dataset, from a local hospital in City D, comprises data from 20 AD patients and 20 age-matched NC volunteers. Each participant provided two EEG samples (eye-closed and eye-open states), recorded using a 16-channel Ag/AgCl electrodes Symtop system at 1024 Hz.

### 4.2 Training Configuration

For each dataset, we created a set of unreliable labels, assigning incorrect labels to a percentage of the samples defined by $\alpha$. Performance was evaluated using true labels, verified, and corrected by experts. Results for $\alpha = 0.3$ are presented as main findings, while those for $\alpha = 0.5$ are detailed in the Appendix. The MACS model was trained over 30 epochs using Stochastic Gradient Descent with a momentum of 0.9 and a weight decay of $10^{-4}$. A step-learning rate scheduler improved training efficiency by starting at 0.1 and reducing it by a factor of 0.1 every 10 epochs. Similarly, a three-fold cross-validation was executed on the PD dataset from the NMU center, with a batch size of 60 We conducted four-fold cross-validation on the MCI dataset with a batch size of 128, and three-fold cross-validation on the PD dataset from the NMU center with a batch size of 60. The MACS framework was developed on a Linux-based system using PyTorch (version 2.0.0), with hardware including a GeForce RTX 3090 GPU and an Intel i9-12900K CPU.

### 4.3 Feature Engineering Study

Feature engineering insights have led to the development of the MACS *Encoder*. EEG frequency markers [7]are inductive for PD but found less effective for MCI. EEG complexity measured by the entropy[8] and Hjorth parameter[6] enhanced MCI detection but was less effective than frequency domain features in PD cases. We compared the functional connectivity characterized by Pearson correlation (Corr) in Euclidean space [41] and SPD in manifold space [36] and found the advantage of manifold representations. This evidence has informed the integration of a specialized block in the *Encoder* to process underlying temporal-spectral-spatial features and analyze synchronization characteristics through manifold geometry.

### 4.4 Comparison Study

We benchmarked MACS against SOTA methods, evaluating its performance in scenarios with unreliable annotations and independently assessing the *Encoder* in scenarios with clean annotations (see Table 2). For the clean data learning comparison, we employed advanced EEG encoders that target manifold geometry[10], integrate attention mechanisms[11], and utilize contrastive learning[3]. Given the absence of a dedicated SOTA model for EEG data with unreliable annotations, we compared MACS against leading methods in the time series[20] and natural image domain[21, 22]. Specifically, we replaced their encoders with our *Encoder* in MACS to validate the efficacy of our combined manifold-Euclidean representation learning strategy for EEG signals. Overall, MACS's weakly supervised learning mechanism, combined with an effective *Encoder*, enables superior comprehensive performance across two disease types. We unveil MACS's learning process through t-SNE mapping in Figure 5(see Appendix for the PD case). Observations indicate that MACS gradually selects the correct trust samples, enhancing representation learning and forming class-specific clusters.

### 4.5 Hyperparameter Tuning Study

*Memory Length.* The multi-view contrastive loss is crucial for the MACS framework. Effective contrastive learning heavily depends on the adequacy of positive and negative pairs; therefore,

**Table 1: Feature engineering assesses the effectiveness of various EEG domains in neurodegeneration recognition.**

| Feature Domains | [MCI] 4-Fold Cross-Validation | | [PD] 3-Fold Cross-Validation | | Comprehensive Average | |
|---|---|---|---|---|---|---|
| | Accuracy | F1 | Accuracy | F1 | Accuracy | F1 |
| Frequency[7] | 58.64(2.06) | 66.03(1.47) | 74.07(0.07) | 71.90(0.02) | 66.36 | 71.51 |
| Entropy[8] | 63.07(1.46) | **69.89(1.50)** | 61.11(0.21) | 57.68(2.29) | 62.09 | 72.37 |
| Hjorth[6] | **67.05(0.80)** | 68.37(0.75) | 64.81(0.27) | 63.78(0.49) | 65.93 | 64.65 |
| Functional Connectivity-Corr[41] | 65.34(1.64) | 67.56(2.58) | 66.67(0.21) | 61.39(0.78) | 66.01 | 63.99 |
| Functional Connectivity-SPD [36] | 64.43(0.77) | 68.18(0.99) | **79.63(0.27)** | **79.46(0.17)** | **72.03** | **74.80** |

**Table 2: Comparative study compares the SOTA methods for learning from both well-annotated and unreliable data.**

| Scenarios | Methods | | [MCI] 4-Fold | | [PD] 3-fold | | Overall Accuracy |
|---|---|---|---|---|---|---|---|
| | | | Accuracy | F1 | Accuracy | F1 | |
| Clean Annotation | [10] [Ju et al., TNNLS, 2022] | Tensor-CSPNet | 80.78(7.93) | 77.70(11.63) | 75.92(6.41) | 73.20(8.16) | 78.35(3.44) |
| | [11] [Pan et al., NeurIPS, 2022] | MAtt | 81.97(3.96) | 79.56(8.29) | 79.63(11.56) | 83.07(7.56) | 80.80(1.65) |
| | [3] [Wang et al., NeurIPS, 2023] | COMET | 73.25(26.02) | 70.09(24.82) | 75.47(9.75) | 74.92(14.17) | 74.36(1.57) |
| | *Encoder* in **MACS** | Final Epoch | **87.65(4.31)** | **86.47(7.73)** | **85.18(8.48)** | **83.46(11.67)** | **86.42(1.75)** |
| Unreliable Annotation | [20][Ma et al., IJCAI, 2023] | CTW | 75.40(10.28) | 76.75(9.80) | 75.93(3.21) | 78.73(2.20) | 75.67(0.37) |
| | [21][Xiao et al., IJCAI, 2023] | Promix | 55.14(18.94) | 56.17(11.49) | 79.63(8.49) | 77.34(10.97) | 67.39(17.32) |
| | [20] [Ma et al., IJCAI, 2023] [21] [Xiao et al., IJCAI, 2023] | CTW Encoder + Promix | 79.75(4.69) | 80.83(7.21) | 81.48(6.41) | 80.25(8.15) | 80.62(1.22) |
| | [22] [Li et al., CVPR, 2022] | Sel-CL | 63.29(22.32) | 54.82(19.42) | 66.67(5.56) | 65.42(4.49) | 64.98(2.39) |
| | [22] [Li et al., CVPR, 2022] | Sel-CL+ | 74.16(7.73) | 73.41(11.98) | 57.41(21.03) | 43.06(37.35) | 65.79(11.84) |
| | [21] [Xiao et al., IJCAI, 2023] | *Encoder* for Promix | 85.38(4.41) | 85.56(4.90) | 83.33(5.56) | 81.05(8.49) | 84.36(1.45) |
| | [22] [Li et al., CVPR, 2022] | *Encoder* for Sel-CL | 72.08(12.76) | 70.05(9.37) | 83.33(5.56) | 81.51(9.04) | 77.71(7.95) |
| | [22] [Li et al., CVPR, 2022] | *Encoder* for Sel-CL+ | 85.43(4.16) | 84.18(7.01) | 83.33(5.56) | 81.05(8.49) | 84.38(1.48) |
| | **MACS** | Final Epoch | **88.74(4.61)** | **88.18(7.23)** | **87.04(3.21)** | **86.40(1.90)** | **87.89(1.20)** |

inspired by [40], we proposed maintaining a memory of previous steps to circumvent the constraints imposed by batch size, enabling more extensive use of available small-scale data. For example, we summarize the research findings for MCI in Table 3, suggesting that memory length contributes to learning efficiency, but its relationship is not monotonic. Please note that the methods involving contrastive learning are discussed in Table 2, where we also determined their optimal performance to ensure a fair comparison.

*Temporal Scale* Temporal scale refers to the time interval for constructing dynamic functional networks in the *Encoder*. The findings indicate that a second-level temporal scale is crucial for capturing patterns. For example, results from the MCI dataset are included in Table 3. More effective EEG markers that appear at larger temporal scales may be attributed to the 'slowing' phenomenon in brain

activity, which is more prominently manifested in patients with neurodegenerative disorders [43, 44].

## 4.6 Ablation Study

Table 4 outlines the impacts of MACS's components, where 'w/o Memory' denotes omitting large-scale storage in contrastive learning. The findings on *Augmentor* highlight the importance of comparing dual brain variants. The variants enhanced by interpolation in *Switcher* significantly benefit small-scale clinical data. The *Stratifier*'s necessity was confirmed by evaluating classification loss using all data in mini-batch without considering confidence levels (denoted as 'w/o Confidence'), and by giving discriminative feedback only for trusted samples, omitting auxiliary input for distrusted samples (denoted as 'w/o Auxiliary'). Furthermore, we implemented cross-entropy loss in the model devoid of confidence stratification, marked as 'w/o CS', to validate our hypothesis about this mechanism's effectiveness.

**Table 3: Investigation of hyper-parameters in MACS.**

| Parameters | Configurations | Accuracy | F1 |
|---|---|---|---|
| Memory Length | 0 | 83.10(6.93) | 83.64(9.09) |
| | 200 | 87.60(4.46) | 86.39(7.58) |
| | 300 | 88.74(4.61) | 88.18(7.23) |
| | 400 | 86.31(5.55) | 84.71(9.69) |
| | 500 | 85.33(4.52) | 84.07(7.77) |
| Temporal Scale | 2s | 88.74(4.61) | 88.18(7.23) |
| | 1s | 87.65(4.31) | 86.75(7.87) |
| | 500ms | 74.06(6.09) | 77.60(8.15) |

**Table 4: Ablation study of the components in MACS.**

| Configuration | [MCI] 4-Fold | | [PD] 3-Fold | |
|---|---|---|---|---|
| | Accuracy | F1 | Accuracy | F1 |
| **MACS** | **88.74(4.61)** | **88.18(7.23)** | **87.04(3.21)** | **86.40(1.90)** |
| w/o Memory | 83.10(6.93) | 83.64(9.09) | 85.18(3.21) | 85.15(3.37) |
| w/o *Augmentor* | 84.24(5.95) | 82.17(11.57) | 85.18(3.21) | 85.32(1.89) |
| w/o *Switcher* | 87.55(6.89) | 86.05(9.79) | 83.33(5.56) | 81.05(8.49) |
| w/o Confidence | 84.19(9.62) | 84.68(12.24) | 79.63(6.41) | 79.46(5.01) |
| w/o Auxiliary | 81.92(7.58) | 82.23(9.72) | 83.33(0.0) | 82.20(2.10) |
| w/o MA | 84.39(7.31) | 83.99(5.98) | 79.63(3.20) | 76.88(6.59) |
| w/o CS | 70.95(10.53) | 68.13(9.39) | 77.78(9.62) | 75.08(13.02) |

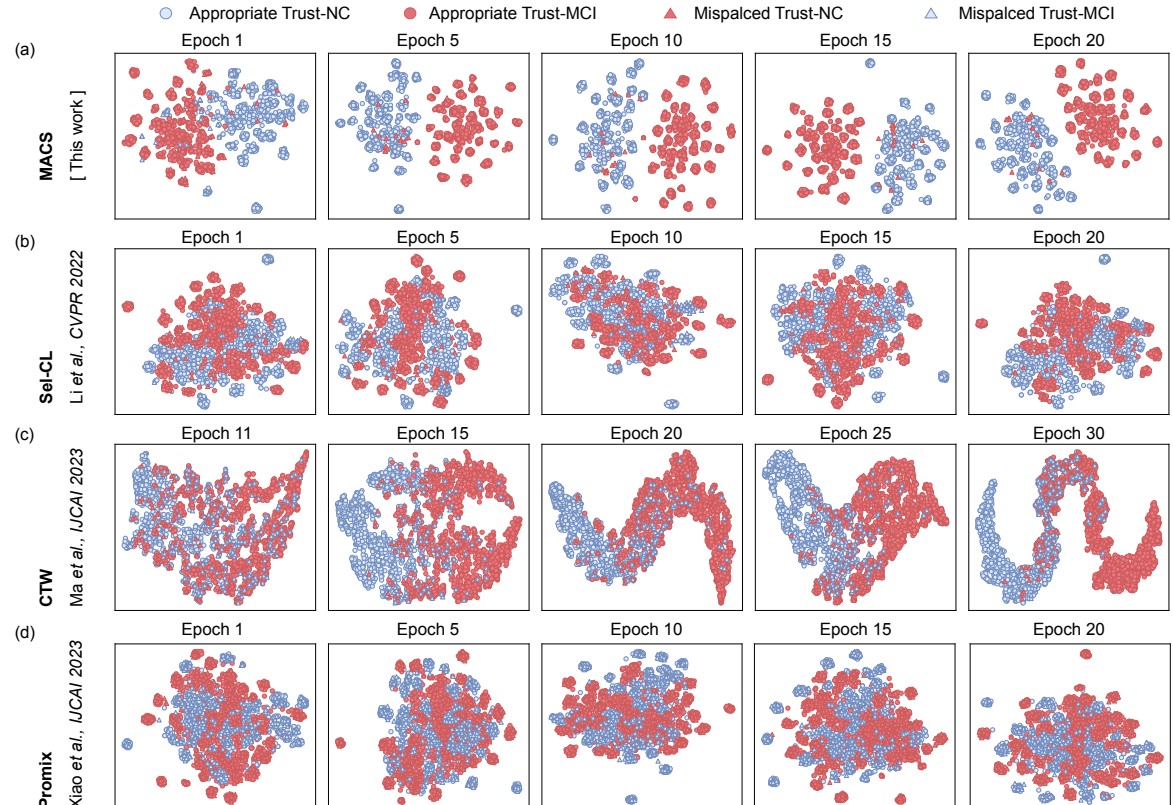

**Figure 5: Qualitative comparison of MACS with SOTA frameworks using t-SNE visualization based on latent distribution.**

## 4.7 Transferability Evaluation

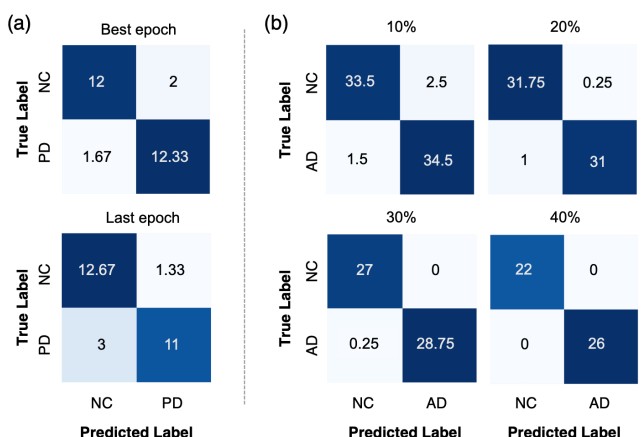

**Figure 6: Evaluation of MACS's Cross-center Transferability: (a) Direct testing results of MACS, trained on Center A's PD data applied to Center B's PD data; (b) Fine-tuning testing results of MACS, trained on Center C's MCI data applied to Center D's AD data, using only a limited percentage of labels.**

From cross-validation tests on PD(NMU) and MCI datasets, we derived N-fold MACS models for PD ($\Theta_{PD}$) and MCI ($\Theta_{MCI}$). We assessed MACS's transferability by testing $\{\theta_{PD}^i\}_{i=0}^2$ on a cross-center PD corpus (from IU), and fine-tuning $\{\theta_{MCI}^i\}_{i=0}^3$ on a cross-center AD corpus. To address the spatial resolution gap between MCI and AD data, we employed EEG's interpolation technique [45]. The averaged evaluation results from $\Theta_{PD}$, halted at both optimal and final epochs (Figure 6(a)), and successful fine-tuning on AD with few labels (Figure 6(b)), underscore MACS's notable transferability in recognizing neurodegenerative disorders via EEG.

## 5 CONCLUSION

This work introduces MACS, a framework that leverages Manifold Attention and Confidence Stratification to address data heterogeneity and annotation unreliability in EEG modeling. Specifically, MACS fuses Euclidean space with manifold geometry to enhance representations and tailors forward and back-propagation based on stratified confidence levels. We have demonstrated superior performance in recognizing two types of diseases and further validated MACS's transferability through cross-center testing and fine-tuning. The techniques used in MACS, effective in signal modeling under unreliable labels, could inspire advances in EEG-based cognitive and emotional computing, as well as in other areas of multimedia representation learning involving low-quality annotations.

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
