# OpenReview forum: "EEG-MACS: Manifold Attention and Confidence Stratification for EEG-based Cross-Center Brain Disease Diagnosis under Unreliable Annotations"
_acmmm.org/ACMMM/2024/Conference — MM2024 Oral_

### Official Review · Reviewer_k3XJ · 2024-05-21

**Rating:** 4
**Confidence:** 2

**Summary:**

This paper introduces the main features and advantages of the MACS framework, including data augmentation, feature space enhancement, an encoder based on Riemannian manifold and Euclidean metrics, a dual-head projector, and sample stratification based on confidence levels. Through these characteristics, the framework has improved performance across different centers and validations, especially in the areas of neurocognitive and movement disorders.

**Strengths:**

The MACS framework proposed in the article has good cross-center data applicability, specifically designed to handle EEG data from different centers, which may vary in collection conditions, equipment, and patient populations.

The framework is capable of processing EEG signals with unreliable annotations, due to its adoption of sample stratification based on confidence levels and self-supervised learning strategies to mitigate the impact of incorrect labels.

**Limitations:**

The specific form of input data is unclear: the EEG sampling equipment and protocols vary across centers, so how many channels are set for the model's input data? What is the sampling rate? What is the data length? For a cross-center model, is it necessary to standardize the number of channels, sampling rate, and data length?

EEG records the noisy electrical activity of a group of neurons, and the authors claim that a temporal scale design of 2 seconds can better capture patterns. Intuitively speaking, a 2-second temporal scale only retains low-frequency information and loses the rich dynamic frequency information of the temporal neural signals.
The article proposes a method for dealing with unreliable annotations, but does not elaborate on how to quantify or assess the unreliability of the annotations.

What does "w/o MA" in Table 4 represent?

It is recommended to modify the author's GIVEN NAME in reference 41 and 44 to non -abbreviation forms.

**Suitability:**

2

---

### Official Review · Reviewer_caAS · 2024-05-21

**Rating:** 5
**Confidence:** 2

**Summary:**

This paper proposes a method for the identification of neurodegenerative diseases, employing Manifold Attention and Confidence Stratification (MACS) to build a transferable network that addresses to some extent the heterogeneity of cross-center data and the unreliability of annotations.

**Strengths:**

1. The overall framework proposed by the article is very innovative, and it is introduced in great detail.
2. The connections and logic among the various modules proposed in the article are very tight, progressively constructing the overall framework.
3. The graphical annotations in the article are rich and clear, facilitating easy reading and understanding.
4. The author conducted thorough comparative experiments and meticulously categorized the main methods used in each comparison experiment.

**Limitations:**

1. The author mentions comparisons with other methods in the Switcher section, but these are not reflected in the experimental section.
2. The author extensively uses dual-branch structures in the framework construction; could this usage adversely affect the model's speed and parallelism?
3. Some expressions in the article are technical and significant, but they lack fluency and readability, such as the ‘the ineffective mapping that cannot project EEG signals into a high-level feature space invariant across diverse data distributions from multiple centers;’ in the Introduction section.

**Suitability:**

2

---

### Official Review · Reviewer_uVnE · 2024-05-26

**Rating:** 6
**Confidence:** 3

**Summary:**

The paper addresses the challenges posed by data heterogeneity and unreliable annotations in the analysis and interpretation of multimedia and multimodal data, particularly in human-centered computing and small-group settings. It focuses on the automated diagnosis of cognitive and movement brain disorders using electroencephalogram (EEG) signals, which is complicated by disease-specific models, single-center studies, and the need for fully supervised learning with annotated data. Authors propose a novel framework called Manifold Attention and Confidence Stratification (MACS), aimed at developing a transferable model that can manage unreliable annotations. The paper demonstrates MACS's superior performance compared to state-of-the-art methods in learning from unreliable annotations, validated across different diseases and through cross-center testing and fine-tuning.

**Strengths:**

The authors make their code publicly available which is of significant broader impact to the community. The authors test their proposed approach on multiple datasets comprehensively and through ablation studies. Moreover, the authors present comparison to existing works for demonstration of superiority of their proposed approach.

**Limitations:**

The readability of the paper could have been significantly improved, which hinders the clarity of parts of the paper significantly. Some parts of algorithm 1 are not clear such as the utility of the augmentor generated samples. The idea that augmentor augments both trusted and untrusted samples needs justifications. There are a number of selections and statements which should be better justified.

**Suitability:**

3

---

### Official Review · Reviewer_xCKg · 2024-05-29

**Rating:** 3
**Confidence:** 2

**Summary:**

This work introduces MACS, a framework that leverages Manifold Attention and Confidence Stratification to address data heterogeneity and annotation unreliability in EEG modeling. Specific advantages and disadvantages are shown below.

**Strengths:**

1. The figure of the article is pleasing to the eye.
2. The structure and logic of the article is clear.
3. The description of the content study is very full.

**Limitations:**

I will raise the score if the questions are answered carefully and satisfactorily.
1. Your article doesn't really have anything to do with multimodality or multimedia, but you make a point of emphasizing the article's relationship to multimedia and multimodality at the end of the abstract and at the beginning of the introduction but never mention it again in the rest of the article, so your mention seems very deliberate, explain it.
2. You mention “the absence of integrating advanced learning strategies to tackle complex learning issues under unreliable annotations effectively.” at the end of the second paragraph of the introduction, which is not a valid reason. Combined with the fact that your approach incorporates multiple modules, this approach makes it look like you're piling up the latest strategies into a single approach, so I can think of your approach as insufficiently innovative.

**Suitability:**

1

---

### Official Review · Reviewer_F9FF · 2024-06-06

**Rating:** 3
**Confidence:** 2

**Summary:**

This paper focuses on the challenges of cross-center data heterogeneity and unreliable annotations in the intelligent diagnosis of diseases using brain signals. It proposes a transferable framework employing manifold attention and confidence stratification to diagnose neurodegenerative disorders based on EEG signals sourced from four centers with unreliable annotations. Experiments verify the effectiveness of the proposed model on EEG-based diagnostics.

**Strengths:**

1. The paper studies the challenging issues of cross-center data heterogeneity and unreliable annotations, which is interesting and contributive to the community.
2. The presentation of the proposed model is clean. It introduces the confidence stratification-based constraintsto enhance representation learning andconfidence assessment, which is interesting.
3. Extensive experiments show the superiority of the proposed model, verifying the effectiveness of the proposed model on EEG-based diagnostics.

**Limitations:**

1. The proposed model is not straightforward or self-motivated. The correspondence between the proposed model and the targeted challenges is not well established.
2. The introduction of the proposed model is complicated and unclear. The time complexity of the proposed model is not explored.
3. The introduction to previous work related to tackling cross-center data heterogeneity is not
previous work related to the
4. Although extensive experiments are performed, it lacks convincing experiments or visualizations to demonstrate the insights behand the model.

**Suitability:**

2

---

### Meta-Review · Area_Chair_XCTv · 2024-06-27

**Recommendation:** Accept (Oral)
**Confidence:** 5

**Metareview:**

This paper explored the automated diagnosis of brain disorders from the perspective of heterogeneity and unreliability issues from EEG data. All reviews are positive after the rebuttal stage, and the authors need to revise the paper and close these concerns in the final version.